# RESTRICTING THE FLOW: INFORMATION BOTTLENECKS FOR ATTRIBUTION

**Karl Schulz**[1*†]**, Leon Sixt**[2*]**, Federico Tombari**[1]**, Tim Landgraf**[2]

* contributed equally   † work done at the Freie Universität Berlin

Technische Universität München[1] Freie Universität Berlin[2]

Corresponding authors: karl.schulz@tum.de, leon.sixt@fu-berlin.de

## ABSTRACT

Attribution methods provide insights into the decision-making of machine learning models like artificial neural networks. For a given input sample, they assign a relevance score to each individual input variable, such as the pixels of an image. In this work, we adopt the information bottleneck concept for attribution. By adding noise to intermediate feature maps, we restrict the flow of information and can quantify (in bits) how much information image regions provide. We compare our method against ten baselines using three different metrics on VGG-16 and ResNet-50, and find that our methods outperform all baselines in five out of six settings. The method's information-theoretic foundation provides an absolute frame of reference for attribution values (bits) and a guarantee that regions scored close to zero are not necessary for the network's decision.

## 1 INTRODUCTION

Deep neural networks have become state of the art in many real-world applications. However, their increasing complexity makes it difficult to explain the model's output. For some applications such as in medical decision making or autonomous driving, model interpretability is an important requirement with legal implications. Attribution methods (Selvaraju et al., 2017; Zeiler & Fergus, 2014; Smilkov et al., 2017) aim to explain the model behavior by assigning a relevance score to each input variable. When applied to images, the relevance scores can be visualized as heatmaps over the input pixel space, thus highlighting salient areas relevant for the network's decision.

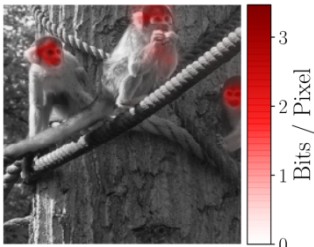

Figure 1: Exemplary heatmap of the Per-Sample Bottleneck for the VGG-16.

For attribution, no ground truth exists. If an attribution heatmap highlights subjectively irrelevant areas, this might correctly reflect the network's unexpected way of processing the data, or the heatmap might be inaccurate (Nie et al., 2018; Viering et al., 2019; Sixt et al., 2019). Given an image of a railway locomotive, the attribution map might highlight the train tracks instead of the train itself. Current attribution methods cannot guarantee that the network is ignroing the low-scored locomotive for the prediction.

We propose a novel attribution method that estimates the amount of information an image region provides to the network's prediction. We use a variational approximation to upper-bound this estimate and therefore, can guarantee that areas with zero bits of information are not necessary for the prediction. Figure 1 shows an exemplary heatmap of our method. Up to 3 bits per pixel are available for regions corresponding to the monkeys' faces, whereas the tree is scored with close to zero bits per pixel. We can thus guarantee that the tree is not necessary for predicting the correct class.

To estimate the amount of information, we adapt the information bottleneck concept (Tishby et al., 2000; Alemi et al., 2017). The bottleneck is inserted into an existing neural network and restricts the information flow by adding noise to the activation maps. Unimportant activations are replaced almost entirely by noise, removing all information for subsequent network layers. We developed two approaches to learn the parameters of the bottleneck – either using a single sample (Per-Sample Bottleneck), or the entire dataset (Readout Bottleneck).

We evaluate against ten different baselines. First, we calculated the Sensitivity-n metric proposed by Ancona et al. (2018). Secondly, we quantified how well the object of interest was localized using bounding boxes and extend the degradation task proposed by Ancona et al. (2017). In all these metrics, our method outperforms the baselines consistently. Additionally, we test the impact of cascading layer-wise weight randomizations on the attribution heatmaps (Adebayo et al., 2018). For reproducibility, we share our source code and provide an easy-to-use[*] implementation.

We name our method *IBA* which stands for Information Bottleneck Attribution. It provides a theoretic upper-bound on the used information while demonstrating strong empirical performance. Our work improves model interpretablility and increases trust in attribution results.

To summarize our contributions:

- We adapt the information bottleneck concept for attribution to estimate the information used by the network. Information theory provides a guarantee that areas scored irrelevant are indeed not necessary for the network's prediction.
- We propose two ways – *Per-Sample* and *Readout* Bottleneck – to learn the parameters of the information bottleneck.
- We contribute a novel evaluation method for attribution based on bounding boxes and we also extend the metric proposed by Ancona et al. (2017) to provide a single scalar value and improve the metric's comparability between different network architectures.

## 2 RELATED WORK

Attribution is an active research topic. *Gradient Maps* (Baehrens et al., 2010) and *Saliency Maps* (Simonyan & Zisserman, 2014) are based on calculating the gradient of the target output neuron w.r.t. to the input features. *Integrated Gradient* (Sundararajan et al., 2017) and *SmoothGrad* (Smilkov et al., 2017) improve over gradient-based attribution maps by averaging the gradient of multiple inputs, either over brightness level interpolations or in a local neighborhood. Other methods, such as *Layerwise Relevance Propagation (LRP)* (Bach et al., 2015), *Deep Taylor Decomposition (DTD)* (Montavon et al., 2017), *Guided Backpropagation (GuidedBP)* (Springenberg et al., 2014) or *DeepLIFT* (Shrikumar et al., 2017) modify the propagation rule. *PatternAttribution* (Kindermans et al., 2018) builds upon DTD by estimating the signal's direction for the backward propagation. Perturbation-based methods are not based on backpropagation and treat the model as a black-box. *Occlusion* (Zeiler & Fergus, 2014) measures the importance as the drop in classification accuracy after replacing individual image patches with zeros. Similarly, blurring image patches has been used to quantify the importance of high-frequency features (Greydanus et al., 2018) .

*Grad-Cam* (Selvaraju et al., 2017) take the activations of the final convolutional layer to compute relevance scores. They also combine their method with GuidedBP: *GuidedGrad-CAM*. Ribeiro et al. (2016) uses image superpixels to explain deep neural networks. High-level concepts rather input pixels are scored by *TCAV* (Kim et al., 2018). Khakzar et al. (2019) prune irrelevant neurons. Similar to our work, Macdonald et al. (2019) uses a rate-distortion perspective, but minimize the norm of the mask instead of the shared information. To the best of our knowledge, we are the first to estimate the amount of used information for attribution purposes.

Although many attribution methods exist, no standard evaluation benchmark is established. Thus, determining the state of the art is difficult. The performance of attribution methods is highly dependent on the used model and dataset. Often only a purely visual comparison is performed (Smilkov et al., 2017; Springenberg et al., 2014; Montavon et al., 2017; Sundararajan et al., 2017; Bach et al., 2015). The most commonly used benchmark is to degrade input images according to the attribution heatmap and measure the impact on the model output. (Kindermans et al., 2018; Samek et al., 2016; Ancona et al., 2017). The Sensitivity-n score (Ancona et al., 2018) is obtained by randomly masking the input image and then measuring the correlation between removed attribution mass and model performance. For the ROAR score (Hooker et al., 2018), the network is trained from scratch on the degraded images. It is computationally expensive but avoids out-of-domain inputs – an inherent problem of masking. Adebayo et al. (2018) proposes a sanity check: if the network's parameters are randomized, the attribution output should change too.

---

[*]`https://github.com/BioroboticsLab/IBA`

Adding noise to a signal reduces the amount of information (Shannon, 1948). It is a popular way to regularize neural networks (Srivastava et al., 2014; Kingma et al., 2015; Gal et al., 2017). For regularization, the noise is applied independently from the input and no attribution maps can be obtained. In Variational Autoencoders (VAEs) (Kingma & Welling, 2013), noise restricts the information capacity of the latent representation. In our work, we construct a similar information bottleneck that can be inserted into an existing network. Deep convolutional neural networks have been augmented with information bottlenecks before to improve the generalization and robustness against adversarial examples (Achille & Soatto, 2018; Alemi et al., 2017). Zhmoginov et al. (2019) and Taghanaki et al. (2019) extract salient regions using information bottlenecks. In contrast to our work, they add the information bottleneck already when training the network and do not focus on post-hoc explanations.

## 3 INFORMATION BOTTLENECK FOR ATTRIBUTION

Instead of a backpropagation approach, we quantify the flow of information through the network in the *forward* pass. Given a pre-trained model, we inject noise into a feature map, which restrains the flow of information through it. We optimize the intensity of the noise to minimize the information flow, while simultaneously maximizing the original model objective, the classification score. The parameters of the original model are not changed.

### 3.1 INFORMATION BOTTLENECK FOR ATTRIBUTION

Generally, the information bottleneck concept (Tishby et al., 2000) describes a limitation of available information. Usually, the labels $Y$ are predicted using all information from the input $X$. The information bottleneck limits the information to predict $Y$ by introducing a new random variable $Z$. The information the new variable $Z$ shares with the labels $Y$ is maximized while minimizing the information $Z$ and $X$ share :

$$\max I[Y; Z] - \beta I[X, Z] , \tag{1}$$

where $I[X, Z]$ denotes the mutual information and $\beta$ controls the trade-off between predicting the labels well and using little information of $X$. A common way to reduce the amount of information is to add noise (Alemi et al., 2017; Kingma & Welling, 2013).

For attribution, we inject an information bottleneck into a pretrained network. The bottleneck is inserted into an layer which still contains local information, e.g. for the ResNet the bottleneck is added after conv3_* (after the last conv3 block). Let $R$ denote the intermediate representations at this specific depth of the network, i.e. $R = f_l(X)$ where $f_l$ is the $l$-th layer output. We want to reduce information in $R$ by adding noise. As the neural network is trained already, adding noise should preserve the variance of the input to the following layers. Therefore, we also damp the signal $R$ when increasing the noise, effectively replacing the signal partly with noise. In the extreme case, when no signal is transmitted, we replace $R$ completely with noise of the same mean and variance as $R$. For this purpose, we estimate the mean $\mu_R$ and variance $\sigma_R^2$ of each feature of $R$ empirically. As information bottleneck, we then apply a linear interpolation between signal and noise:

$$Z = \lambda(X)R + (1 - \lambda(X)) \epsilon , \tag{2}$$

where $\epsilon \sim \mathcal{N}(\mu_R, \sigma_R^2)$ and $\lambda(X)$ controls the damping of the signal and the addition of the noise. The value of $\lambda$ is a tensor with the same dimensions as $R$ and $\lambda_i \in [0, 1]$. Given $\lambda_i(X) = 1$ at the feature map location $i$, the bottleneck transmits all information as $Z_i = R_i$. Whereas if $\lambda_i = 0$, then $Z_i = \epsilon$ and all information of $R_i$ is lost and replaced with noise. It could be tempting to think that $Z$ from equation 2 has the same mean and variance as $R$. This is not the case in general as $\lambda(X)$ and $R$ both depend on $X$ (for more detail see Appendix E).

In our method, we consider an area relevant if it contains useful information for classification. We need to estimate how much information $Z$ still contains about $R$. This quantity is the mutual information $I[R, Z]$ that can be written as:

$$I[R, Z] = \mathbb{E}_R[D_{\mathrm{KL}}[P(Z|R)||P(Z)]] , \tag{3}$$

where $P(Z|R)$ and $P(Z)$ denote the respective probability distributions. We have no analytic expression for $P(Z)$ since it would be necessary to integrate over the feature map $p(z) = \int_R p(z|r)p(r)\mathrm{d}r$

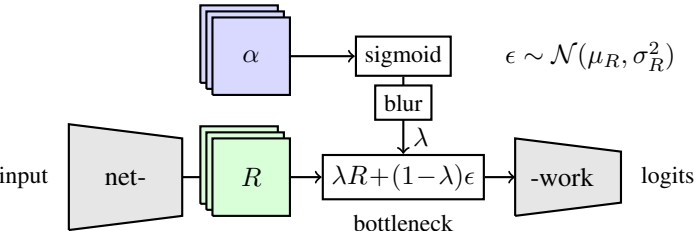

Figure 2: *Per-Sample Bottleneck*: The mask (blue) contains an $\alpha_i$ for each $r_i$ in the intermediate feature maps $R$ (green). The parameter $\alpha$ controls how much information is passed to the next layer. The mask $\alpha$ is optimized for each sample individually according to equation 6.

– an intractable integral. It is a common problem that the mutual information can not be computed exactly but is instead approximated (Poole et al., 2019; Suzuki et al., 2008). We resort to a variational approximation $Q(Z) = \mathcal{N}(\mu_R, \sigma_R)$ which assumes that all dimensions of $Z$ are distributed normally and independent – a reasonable assumption, as activations after linear or convolutional layers tend to have a Gaussian distribution (Klambauer et al., 2017). The independence assumption will generally not hold. However, this only overestimates the mutual information as shown below. Substituting $Q(Z)$ into the previous equation 3, we obtain:

$$\mathrm{I}[R, Z] = \mathbb{E}_R[D_{\mathrm{KL}}[P(Z|R)||Q(Z)]] - D_{\mathrm{KL}}[P(Z)||Q(Z)]. \tag{4}$$

The derivation is shown in appendix D and follows Alemi et al. (2017). The first term contains the KL-divergence between two normal distributions, which is easy to evaluated and we use it to approximate the mutual information. The information loss function $\mathcal{L}_I$ is therefore:

$$\mathcal{L}_I = \mathbb{E}_R[D_{\mathrm{KL}}[P(Z|R)||Q(Z)]]. \tag{5}$$

We know that $\mathcal{L}_I$ overestimates the mutual information, i.e. $\mathcal{L}_I \geq \mathrm{I}[R, Z]$ as the second KL-divergence term $D_{\mathrm{KL}}[P(Z)||Q(Z)]$ has to be positive. If $\mathcal{L}_I$ is zero for an area, we can guarantee that information from this area is not necessary for the network's prediction. Information from this area might still be used when no noise is added.

We aim to keep only the information necessary for correct classification. Thus, the mutual information should be minimal while the classification score should remain high. Let $\mathcal{L}_{CE}$ be the cross-entropy of the classification. Then, we obtain the following optimization problem:

$$\mathcal{L} = \mathcal{L}_{CE} + \beta \mathcal{L}_I, \tag{6}$$

where the parameter $\beta$ controls the relative importance of both objectives. For a small $\beta$, more bits of information are flowing and less for a higher $\beta$. We propose two ways of finding the parameters $\lambda$ – the Per-Sample Bottleneck and the Readout Bottleneck. For the Per-Sample Bottleneck, we optimize $\lambda$ for each image individually, whereas in the readout bottleneck, we train a distinct neural network to predict $\lambda$.

## 3.2    Per-Sample Bottleneck

For the *Per-Sample Bottleneck*, we use the bottleneck formulation described above and optimize $\mathcal{L}$ for individual samples – not for the complete dataset at once. Given a sample $x$, $\lambda$ is fitted to the sample to reflect important and unimportant regions in the feature space. A diagram of the Per-Sample Bottleneck is shown in Figure 2.

**Parameterization**    The bottleneck parameters $\lambda$ have to be in $[0, 1]$. To simplify optimization, we parametrize $\lambda = \mathrm{sigmoid}(\alpha)$. This parametrization allows $\alpha \in \mathbb{R}^d$ and avoids any clipping of $\lambda$ to $[0, 1]$ during optimization.

**Initialization**    As when training neural networks, the initialization of parameters matters. In the beginning, we want to retain all the information. For all dimensions $i$, we initialize $\alpha_i = 5$ and thus $\lambda_i \approx 0.993 \Rightarrow Z \approx R$. At first, the bottleneck has practically no impact on the model performance. It then deviates from this starting point to suppress unimportant regions.

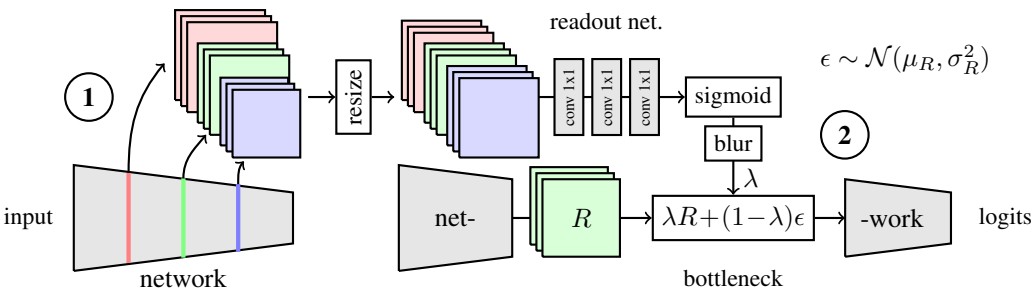

Figure 3: *Readout Bottleneck*: In the first forward pass ①, feature maps are collected at different depths. The readout network uses a resized version of the feature maps to predict the parameters for the bottleneck layer. In the second forward pass ②, the bottleneck is inserted and noise added. All parameters of the analyzed network are kept fixed.

**Optimization** We do 10 iterations using the Adam optimizer (Kingma & Ba, 2014) with learning rate 1 to fit the mask $\alpha$. To stabilize the training, we copy the single sample 10 times and apply different noise to each. In total, we execute the model on 100 samples to create a heatmap, comparable to other methods such as SmoothGrad. After the optimization, the model usually predicts the target with probability close to 1, indicating that only negative evidence was removed.

**Measure of information** To measure the importance of each feature in $Z$, we evaluate $D_{KL}(P(Z|R)||Q(Z))$ per dimension. It shows where the information flows. We obtain a two-dimensional heatmap $m$ by summing over the channel axis: $m_{[h,w]} = \sum_{i=0}^{c} D_{KL}(P(Z_{[i,h,w]}|R_{[i,h,w]})||Q(Z_{[i,h,w]}))$. As convolutional neural networks preserve the locality in their channel maps, we use bilinear interpolation to resize the map to the image size. For the ResNet-50, we insert the bottleneck after layer conv3_*. Choosing a later layer with a lower spatial resolution would increase the blurriness of the attribution maps, due to the required interpolation. The effect of different depth choices for the bottleneck is visualized in figure 4.

**Enforcing local smoothness** Pooling operations and convolutional layers with stride greater than 1 are ignoring parts of the input. The ignored parts cause the Per-Sample Bottleneck to overfit to a grid structure (shown in Appendix B). To obtain a robust and smooth attribution map, we convolve the sigmoid output with a fixed Gaussian kernel with standard deviation $\sigma_s$. Smoothing the mask during training is *not* equivalent to smoothing the resulting attribution map, as during training also the gradient is averaged locally. Combining everything, the parametrization for the Per-Sample Bottleneck is:

$$\lambda = \text{blur}(\sigma_s, \text{sigmoid}(\alpha)) . \tag{7}$$

## 3.3 READOUT BOTTLENECK

For the *Readout Bottleneck*, we train a second neural network to predict the mask $\alpha$. In contrast to the Per-Sample Bottleneck, this model is trained on the entire training set. In Figure 3, the Readout Bottleneck is depicted. Kümmerer et al. (2014) introduced the readout concept for gaze prediction. The general idea is to collect feature maps from different depths and then combine them using 1x1 convolutions.

In a first forward pass, no noise is added and we collect the different feature maps. As the spatial resolution of the feature maps differs, we interpolate them bilinearly to match the spatial dimensions of the bottleneck layer. The readout network then predicts the information mask based on the collected feature maps. In a second forward pass, we insert the bottleneck layer into the network and restrict the flow of information.

Except for the learned importance values, the Readout Bottleneck is identical to the mechanism of the Per-Sample Bottleneck. The measure of information works in the same way as for the Per-Sample Bottleneck and we also use the same smoothing. Given a new sample, we can obtain a heatmap by merely collecting the feature maps and executing the readout network.

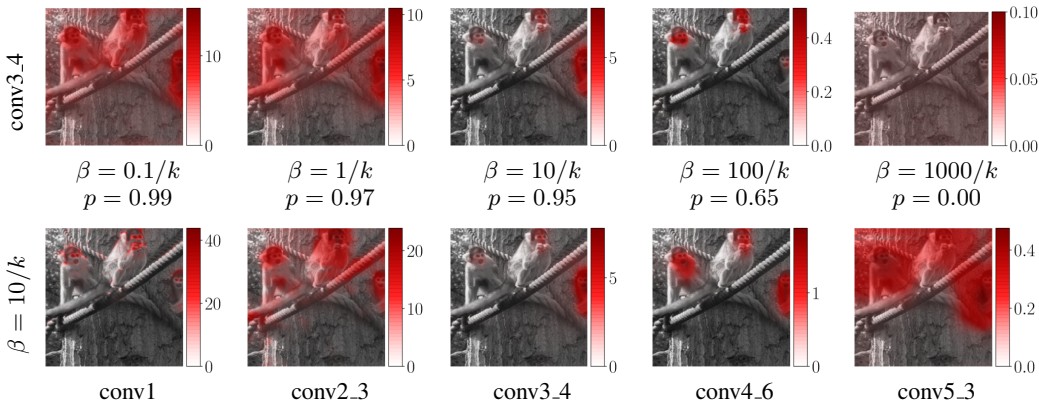

Figure 4: Effect of varying layer depth and $\beta$ on the Per-Sample Bottleneck for the ResNet-50. The color bars measure the bits per input pixel. The resulting output probability of the correct class $p = p(y|x, \beta)$ is decreasing for higher $\beta$.

The readout network consists of three 1x1 convolutional layers. ReLU activations are applied between convolutional layers and a final sigmoid activation yields $\lambda \in [0, 1]$. As the input consists of upscaled feature maps, the field-of-view is large, although the network itself only uses 1x1 conv. kernels.

## 4 EVALUATION

### 4.1 EXPERIMENTAL SETUP

As neural network architectures, we selected the ResNet-50 (He et al., 2016) and the VGG-16 Simonyan & Zisserman (2014), using pretrained weights from the torchvision package (Paszke et al., 2017). These two models cover a range of concepts: dimensionality by stride or max-pooling, residual connections, batch normalization, dropout, low depth (16-weight-layer VGG), and high depth (50-weight-layer ResNet). This variety makes the evaluation less likely to overfit on a specific model type. They are commonly used in the literature concerning attribution methods. For PatternAttribution on the VGG-16, we obtained weights for the signal estimators from Kindermans et al. (2018).

As naive baselines, we selected random attribution, Occlusion with patch sizes 8x8 and 14x14, and Gradient Maps. SmoothGrad and Integrated Gradients cover methods that accumulate gradients. We include three methods with modified backpropagation rules: PatternAttribution, GuidedBP, and LRP. As our implementation of PatternAttribution and LRP does not support skip connections, we report no results for them on the ResNet-50. We also include Grad-CAM and its combination with GuidedBP, GuidedGrad-CAM.

For the compared methods, we use the hyperparameters suggested by the original authors. For LRP (Bach et al., 2015), different rules exist. We include the most commonly used $\alpha$=1, $\beta$=0, $\epsilon$=0 rule which is also displayed in the figures. We also include $\alpha$=0, $\beta$=-1, $\epsilon$=5 as it gives better results on the bounding-box task. For sensitivity-n and sanity checks, only the later LRP variante is evaluted. For our methods, the hyperparameters are obtained using grid search with the degradation metric as objective (Appendix F). The readout network is trained on the training set of the ILSVRC12 dataset (Russakovsky et al., 2015) for $E = 20$ epochs.

The optimization objective of the bottleneck is $\mathcal{L}_{CE} + \beta \mathcal{L}_I$ as given in equation 6. Generally, the information loss $\mathcal{L}_I$ is larger than the classifier loss by several orders of magnitude as it sums over height $h$, width $w$ and channels $c$. We therefore use $k = hwc$ as a reference point to select $\beta$. In figure 4, we display the effect of different orders of $\beta$ and the effect of layer depth for ResNet-50 (see Appendix C for VGG-16). A uniform uninformative heatmap is obtained for $\beta \geq 1000/k$ – all information gets discarded, resulting in an output probability of 0 for the correct class. The heatmap for $\beta = 0.1/k$ is similar to $\beta = 1/k$ but more information is passed, i.e. less noises is added. In appendix F, we also compare pre- to post-training accuracy and estimate the mutual information for

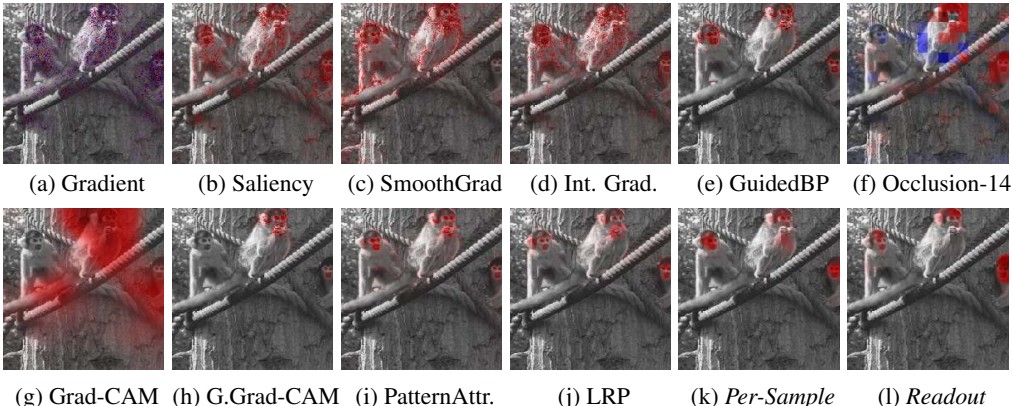

| (a) Gradient | (b) Saliency | (c) SmoothGrad | (d) Int. Grad. | (e) GuidedBP | (f) Occlusion-14 |
|---|---|---|---|---|---|
| (g) Grad-CAM | (h) G.Grad-CAM | (i) PatternAttr. | (j) LRP | (k) *Per-Sample* | (l) *Readout* |

Figure 5: Heatmaps of all implemented methods for the VGG-16 (see Appendix A for more).

both Bottleneck types. For the Per-Sample Bottleneck, we investigate $\beta = 1/k, 10/k, 100/k$. The Readout network is trained with the best performing value $\beta = 10/k$.

## 4.2 QUALITATIVE ASSESSMENT

In Figure 5, the heatmaps of all evaluated samples are shown (for more samples, see Appendix A). Subjectively, both the Per-Sample and Readout Bottleneck identify areas relevant to the classification well. While Guided Backpropagation and PatternAttribution tend to highlight edges, the Per-Sample Bottleneck focuses on image regions. For the Readout Bottleneck, the attribution is concentrated a little more on the object edges. Compared to Grad-CAM, both our methods are more specific, i.e. fewer pixels are scored high.

## 4.3 SANITY CHECK: RANDOMIZATION OF MODEL PARAMETERS

Adebayo et al. (2018) investigates the effect of parameter randomization on attribution masks. A sound attribution method should depend on the entire network's parameter set. Starting from the last layer, an increasing proportion of the network parameters is re-initialized until all parameters are random. We excluded PatternAttribution as the randomization would require to re-estimate the signal directions. The difference between the original heatmap and the heatmap obtained from the randomized model is quantified using SSIM (Wang et al., 2004). We discuss implementation details in the appendix G.1.

In figure 6, we display the results of the sanity check. For our methods, we observe that randomizing the final dense layer drops the mean SSIM by around 0.4. The values for the Readout Bottleneck are of limited expressiveness as we did not re-train it after randomization. For SmoothGrad and Int. Gradients, the SSIM drops by more than 0.4. The heatmaps of GuidedBP and LRP remain similar even if large portion of the network's parameters are randomized – they do not explain the network prediction faithfully. Nie et al. (2018) provides theoretical insights about why GuidedBP fails. Sixt et al. (2019) analyzes why LRP and other modified BP methods fail.

## 4.4 SENSITIVITY-N

Ancona et al. (2018) proposed Sensitivity-n as an evaluation metric for attribution methods. Sensitivity-n masks the network's input randomly and then measures how strongly the amount of attribution in the mask correlates with the drop in classifier score. Given a set $T_n$ containing $n$ randomly selected pixel indices, Sensitivity-n measures the Pearson correlation coefficient:

$$\text{corr}\left(\sum_{i \in T_n} R_i(x), \ S_c(x) - S_c(x_{[x_{T_n}=0]})\right), \tag{8}$$

where $S_c(x)$ is the classifier logit output for class $c$, $R_i$ is the relevance at pixel $i$ and $x_{[x_{T_n}=0]}$ denotes the input with all pixels in $T_n$ set to zero. As in the original paper, we pick the number of masked

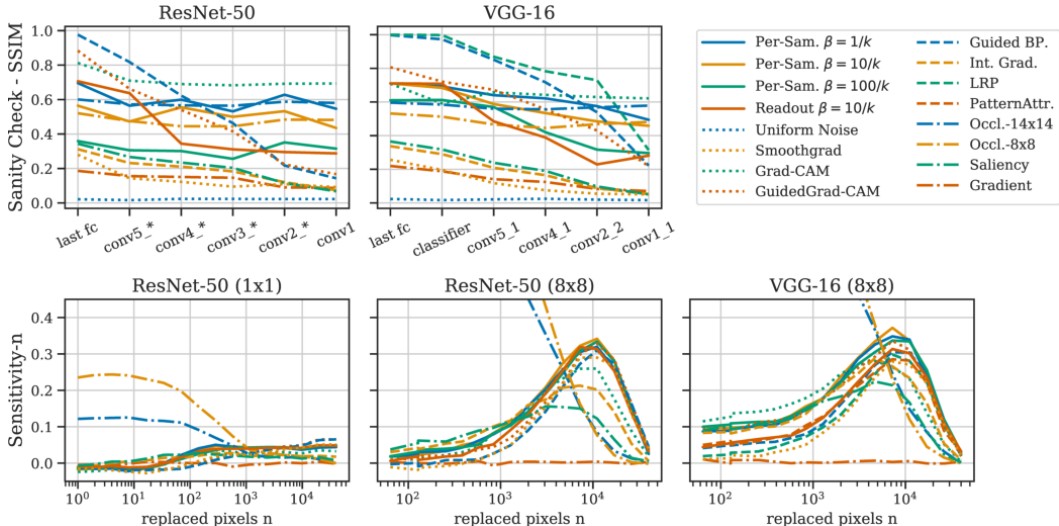

Figure 6: *First row* drop in SSIM score when network layers are randomized. Best viewed in color. *Second-row* Sensitivity-n scores for ResNet-50 and VGG-16. For the ResNet-50, also tile size 1x1 is shown. We clipped the y-axis at 0.4 to improve discriminability.

pixels $n$ in a log-scale between 1 and 80% of all pixels. For each $n$, we generate 100 different index sets $T$ and test each on 1000 randomly selected images from the validation set. The correlation is calculated over the different index sets and then averaged over all images.

In Figure 6, the Sensitivity-n scores are shown for ResNet-50 and VGG-16. When masking inputs pixel-wise as done in Ancona et al. (2018), the Sensitivity-n score across methods are not well discriminable, all scores range within the lower 10% of the scale. We ran the metric with a tile size of 8x8 pixels. This resulted in an increase of the Sensitivity-n score to 30% of the scale. Although not shown in the figure due to the zooming, Occlusion-8x8 performs perfectly for small $n$ as its relevance scores correspond directly to the drop in logits per 8x8 tile. We find that the Per-Sample Bottlenecks $\beta = 10/k$ perform best for both models above $n = 2 \cdot 10^3$ pixels, i.e. when more then 2% of all pixels are masked.

## 4.5 BOUNDING BOX

To quantify how well attribution methods identify and localize the object of interest, we rely on bounding boxes available for the ImageNet dataset. Bounding boxes may contain irrelevant areas, in particular for non-rectangular objects. We restrict our evaluation to images with bounding boxes covering less than 33% of the input image. In total, we run the bounding box evaluation on 11,849 images from the ImageNet validation set.

If the bounding box contains $n$ pixels, we measure how many of the $n$-th highest scored pixels are contained in the bounding box. By dividing by $n$, we obtain a ratio between 0 and 1. The results are shown in Table 1. The Per-Sample Bottleneck outperforms all baselines on both VGG-16 and ResNet-50 each by a margin of 0.152.

An alternative metric would be to take the sum of attribution in the bounding box and compare it to the total attribution in the image. We found this metric is not robust against extreme values. For the ResNet-50, we found basic Gradient Maps to be the best method as a few pixels receiving extreme scores are enough to dominate the sum.

## 4.6 IMAGE DEGRADATION

As a further quantitative evaluation, we rely on the degradation task as used by Ancona et al. (2017); Kindermans et al. (2018); Hooker et al. (2018); Samek et al. (2016). Given an attribution heatmap, the input is split in tiles, which are ranked by the sum of attribution values within each corresponding

| Model & Evaluation | ResNet-50 deg. | | VGG-16 deg. | | ResNet | VGG |
| | 8x8 | 14x14 | 8x8 | 14x14 | bbox | bbox |
|---|---|---|---|---|---|---|
| Random | 0.000 | 0.000 | 0.000 | 0.000 | 0.167 | 0.167 |
| Occlusion-8x8 | 0.162 | 0.130 | 0.267 | 0.258 | 0.296 | 0.312 |
| Occlusion-14x14 | 0.228 | 0.231 | 0.402 | 0.404 | 0.341 | 0.358 |
| Gradient | 0.002 | 0.005 | 0.001 | 0.005 | 0.259 | 0.276 |
| Saliency | 0.287 | 0.305 | 0.326 | 0.362 | 0.363 | 0.393 |
| GuidedBP | 0.491 | 0.515 | 0.460 | 0.493 | 0.388 | 0.373 |
| PatternAttribution | – | – | 0.440 | 0.457 | – | 0.404 |
| LRP $\alpha=1, \beta=0$ | – | – | 0.471 | 0.486 | – | 0.397 |
| LRP $\alpha=0, \beta=1, \epsilon=5$ | – | – | 0.462 | 0.467 | – | 0.441 |
| Int. Grad. | 0.401 | 0.424 | 0.420 | 0.453 | 0.372 | 0.396 |
| SmoothGrad | 0.485 | 0.502 | 0.438 | 0.455 | 0.439 | 0.399 |
| Grad-CAM | 0.536 | 0.541 | 0.510 | 0.517 | 0.465 | 0.399 |
| GuidedGrad-CAM | 0.565 | **0.577** | 0.555 | 0.576 | 0.468 | 0.419 |
| IBA Per-Sample $\beta=1/k$ | **0.573** | 0.573 | 0.581 | 0.583 | 0.606 | 0.566 |
| IBA Per-Sample $\beta=10/k$ | 0.572 | 0.571 | **0.582** | **0.585** | **0.620** | **0.593** |
| IBA Per-Sample $\beta=100/k$ | 0.534 | 0.535 | 0.542 | 0.545 | 0.574 | 0.568 |
| IBA Readout $\beta=10/k$ | 0.536 | 0.536 | 0.490 | 0.536 | 0.484 | 0.437 |

Table 1: *Degradation (deg.)*: Integral between LeRF and MoRF in the degradation benchmark for different models and window sizes over the ImageNet test set. *Bounding Box (bbox)*: the ratio of the highest scored pixels within the bounding box. For ResNet-50, we show no results for PatternAttribution and LRP as no PyTorch implementation supports skip-connections.

tile of the attribution. At each iteration, the highest-ranked tile is replaced with a constant value, the modified input is fed through the network, and the resulting drop in target class score is measured. A steep descent of the accuracy curve indicates a meaningful attribution map.

When implemented in the described way, the most relevant tiles are removed first (MoRF). However, Ancona et al. (2017) argues that using only the MoRF curve for evaluation is not sufficient. For the MoRF score, it is beneficial to find tiles disrupting the output of the neural network as quickly as possible. Neural networks are sensitive to subtle changes in the input (Szegedy et al., 2013). The tiles do not necessarily have to contain meaningful information to disrupt the network. Ancona et al. (2017) proposes to invert the degradation direction, removing tiles ranked as least relevant by the attribution method first (LeRF). The LeRF task favors methods that identify areas sufficient for classification. We scale the network's output probabilities to be in $[0,1]$:

$$s(x) = \frac{p(y|x) - b}{t_1 - b} \, , \qquad (9)$$

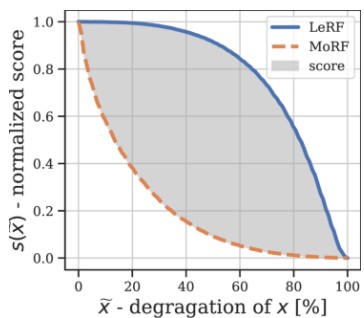

Figure 7: Mean MoRF and LeRF for the Per-Sample Bottleneck. The area is the final degradation score.

where $t_1$ is the model's average top-1 probability on the original samples and $b$ is the mean model output on the fully degraded images. Both averages are taken over the validation set. A score of 1 corresponds to the original model performance.

Both LeRF and MoRF degradation yield curves as visualized in Figure 7, measuring different qualities of the attribution method. To obtain a scalar measure of attribution performance and to combine both metrics, we propose to calculate the integral between the MoRF and LeRF curves.

The results for all implemented attribution methods on the degradation task are given in Table 1. We evaluated both models on the full validation set using 8x8 and 14x14 tiles. In Appendix H, we show the mean LeRF and MoRF curves for 14x14 tiles. The Per-Sample Bottleneck outperforms all other methods in the degradation benchmark except for GuidedGrad-CAM on ResNet-50 where it scores comparably (score difference of $0.004$). The Readout Bottleneck generally achieves a lower degradation scores but still perform competitively.

## 5 CONCLUSION

We propose two novel attribution methods that return an upper bound on the amount of information each input region provides for the network's decision. Our models' core functionality is a bottleneck layer used to inject noise into a given feature layer and a mechanism to learn the parametrized amount of noise per feature. The Per-Sample Bottleneck is optimized per single data point, whereas the Readout Bottleneck is trained on the entire dataset.

Our method does not constrain the internal network structure. In contrast to several modified backpropagation methods, it supports any activation function and network architecture. To evaluate our method, we extended the degradation task to quantify model performance deterioration when removing both relevant and irrelevant image tiles first. We also show results on how well ground-truth bounding boxes are scored. Our Per-Sample and Readout Bottleneck both show competitive results on all metrics used, outperforming state of the art with a significant margin for some of the tasks.

Generally, we would advise using the Per-Sample Bottleneck over the Readout Bottleneck. It performs better and is more flexible as it only requires to estimate the mean and variance of the feature map. The Readout Bottleneck has the advantage of producing attribution maps with a single forward pass once trained. Images with multiple object instances provide the network with redundant class information. The Per-Sample Bottleneck may discard some of the class evidence. Even for single object instances, the heatmaps of the Per-Sample Bottleneck may vary slightly due to the randomness of the optimization process.

The method's information-theoretic foundation provides a guarantee that the network does not require regions of zero-valued attribution for correct classification. To our knowledge, our attribution method is the only one to provide scores with units (bits). This absolute frame of reference allows a quantitative comparison between models, inputs, and input regions. We hope this contributes to a deeper understanding of neural networks and creates trust to use modern models in sensitive areas of application.

ACKNOWLEDGEMENT

We want to thank our anonymous reviewers for their valuable suggestions. In particular, we thank reviewer 1 for encouraging us to include the sanity checks. We are indebted to B. Wild, A. Elimelech, M. Granz, and D. Dormagen for helpful discussions and feedback. For LRP, we used the open source implementation by M. Böhle and F. Eitel (Böhle et al., 2019) and we thank A-K. Dombrowski for sharing an implementation and patterns for PatternAttribution with us. We thank Chanakya Ajit Ekbote for pointing out a mistake in equation 4. LS was supported by the Elsa-Neumann-Scholarship by the state of Berlin. We are also grateful to Nvidia for providing us with a Titan Xp and to ZEDAT for granting us access to their HPC system.

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

# A  VISUAL COMPARISON OF ATTRIBUTION METHODS

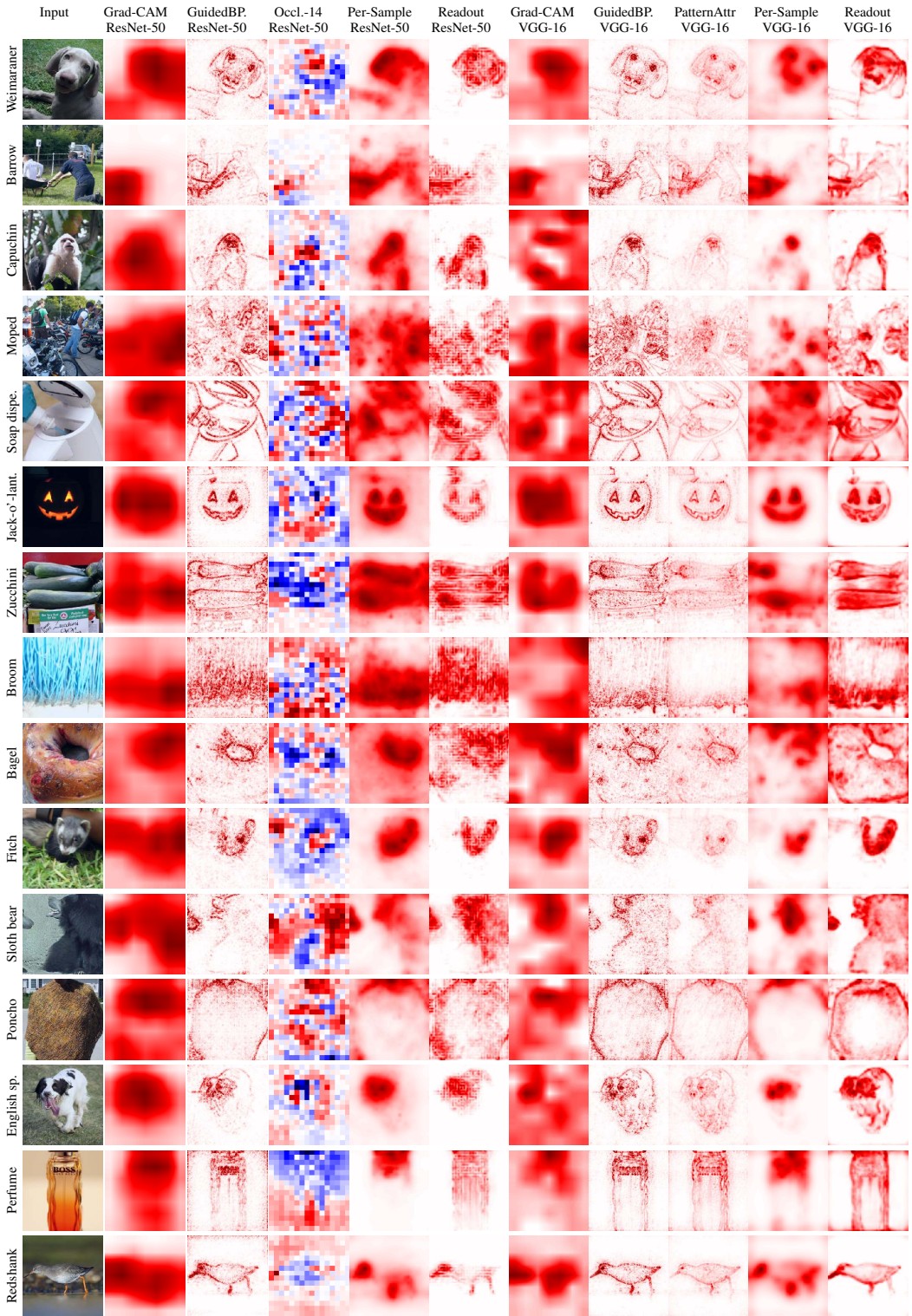

Figure 8: Blue indicates negative relevance and red positive. The authors promise that the samples were picked truly randomly, no cherry-picking, no lets-sample-again-does-not-look-nice-enough.

## B   GRID ARTIFACTS WHEN NOT USING SMOOTHING

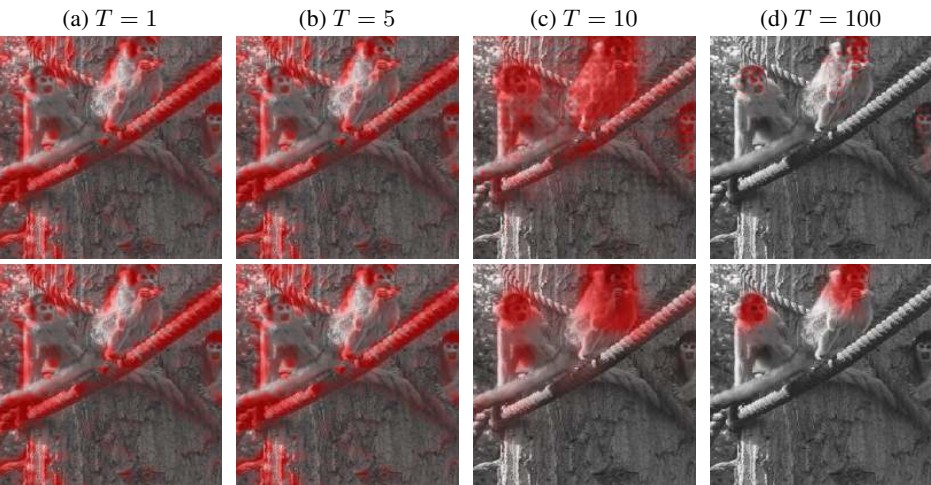

Figure 9: Development of $D_{\mathrm{KL}}(Q(Z|X)||Q(Z))$ of the Per-Sample Bottleneck for layer `conv1_3` of the ResNet-50. Red indicate areas with maximal information flow and semi-transparent green for zero information flow. Top row: without smoothing the mask exhibits a grid structure. Bottom row: smoothing with $\sigma_s = 2$. The smoothing both prevents the artifacts and reduces overfitting to small areas.

## C   EFFECTS OF DIFFERENT $\beta$ AND LAYER DEPTH FOR THE VGG-16

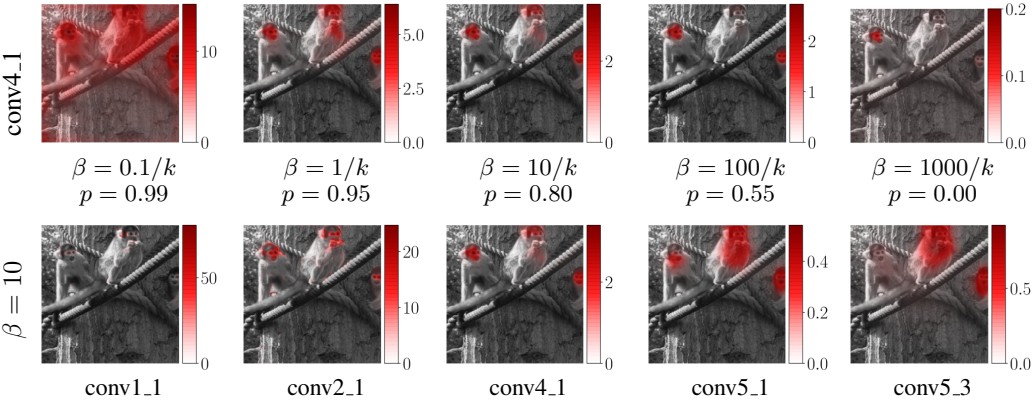

Figure 10: Effect of varying layer depth and $\beta$ on the Per-Sample Bottleneck for the VGG-16. The resulting ouput probability for the correct class is given as $p$.

## D    DERIVATION OF THE UPPER-BOUND OF MUTUAL INFORMATION

For the mutual information $I[R, Z]$, we have:

$$I[R, Z] = \mathbb{E}_R[D_{\text{KL}}[P(Z|R)||P(Z)]] \tag{10}$$

$$= \int_R p(r) \left( \int_Z p(z|r) \log \frac{p(z|r)}{p(z)} dz \right) dr \tag{11}$$

$$= \int_R \int_Z p(r, z) \log \frac{p(z|r)}{p(z)} \frac{q(z)}{q(z)} dz dr \tag{12}$$

$$= \int_R \int_Z p(r, z) \log \frac{p(z|r)}{q(z)} dz dr + \int_R \int_Z p(r, z) \log \frac{q(z)}{p(z)} dz dr \tag{13}$$

$$= \int_R \int_Z p(r, z) \log \frac{p(z|r)}{q(z)} dz dr + \int_Z p(z) \left( \int_R p(r|z) dr \right) \log \frac{q(z)}{p(z)} dz \tag{14}$$

$$= \mathbb{E}_R \left[ D_{\text{KL}}[P(Z|R)||Q(Z)] \right] - D_{\text{KL}}[P(Z)||Q(Z)] \tag{15}$$

$$\leq \mathbb{E}_R \left[ D_{\text{KL}}[P(Z|R)||Q(Z)] \right] \tag{16}$$

## E    MEAN AND VARIANCE OF $Z$

The $\lambda(X)$ linearly interpolate between the feature map $R$ and the noise $\epsilon \sim \mathcal{N}(\mu_R, \sigma_R^2)$, where $\mu_R$ and $\sigma_R$ are the estimated mean and standard derivation of $R$. Both $R = f_l(X)$ and $\lambda(X)$ depend on the input random variable $X$.

$$Z = \lambda(X)R + (1 - \lambda(X))\epsilon \tag{17}$$

For the mean of $Z$, we have:

$$
\begin{aligned}
\mathbb{E}[Z] &= \mathbb{E}[\lambda(X)R] + \mathbb{E}[(1 - \lambda(X))\epsilon] && \triangleright \text{substituting in definition of } Z \\
&= E[\lambda(X)R] + \mathbb{E}[1 - \lambda(X)]\mathbb{E}[\epsilon] && \triangleright \text{independence of } \lambda \text{ and } \epsilon \\
&= \text{cov}(\lambda(X), R) + \mathbb{E}[\lambda(X)]\mathbb{E}[R] + \mathbb{E}[1 - \lambda(X)]\mathbb{E}[\epsilon] && \triangleright \text{cov}(A, B) = \mathbb{E}[AB] - \mathbb{E}[A]\mathbb{E}[B] \\
&\approx \text{cov}(\lambda(X), R) + \mathbb{E}[\epsilon] && \triangleright \mathbb{E}[\epsilon] = \mu_R \approx \mathbb{E}[R]
\end{aligned}
$$

As $\lambda(X)$ and $R$ are multiplied together, they form a complex product distribution. If they do not correlate, $\mathbb{E}[Z] \approx \mathbb{E}[\epsilon] \approx \mathbb{E}[R]$.

A similar problem araises for the variance:

$$
\begin{aligned}
\text{Var}[Z] &= \mathbb{E}[Z^2] - \mathbb{E}[Z]^2 \\
&= \mathbb{E}[(\lambda(X)R + (1 - \lambda(X)\epsilon)^2] - (\text{cov}(\lambda(X), R) + \mathbb{E}[\epsilon])^2
\end{aligned}
$$

The multiplication of $\lambda(X)$ and $R$ causes in general the variance of $Z$ and $R$ to not match: $\text{Var}[Z] \neq \text{Var}[R]$.

## F    HYPERPARAMETERS

| Parameter | ResNet-50 | VGG-16 | Search space |
|---|---|---|---|
| Target layer | `conv3_4` | `conv4_1` | |
| Optimizer | Adam (Kingma & Ba (2014)) | | |
| Learning Rate | $\eta = 1$ | | $\{0.03, 0.1, 0.3, 1, 3, 10\}$ |
| Balance Factor | $\beta = 10/k$ | | $\{0.001, 0.01, 0.1, 1, 10, 100, 300\}$ |
| Iterations | $T = 10$ | | $\{1, 3, 5, 10, 30, 100\}$ |
| Batch Size | $B = 10$ | | $\{1, 5, 10, 30\}$ |
| Smoothing | $\sigma_s = 1$ | | $\{0.5, 1, 2\}$ |

Table 2: Hyperparameters for Per-Sample Bottleneck. The layer notations for the ResNet-50 are taken from the original publication (He et al., 2016). The first index denotes the block and the second the layer within the block. For the VGG-16, `conv_n` denotes the n-th convolutional layer.

|  |  | Orig. | Initial | $\beta=0.01/k$ | $0.1/k$ | $1/k$ | $10/k$ | $100/k$ |
|---|---|---|---|---|---|---|---|---|
| Per-Sample | $\mathcal{L}_I/k$ | – | 2.500 | 3.174 | 0.525 | 0.248 | 0.098 | 0.019 |
|  | Top-5 Acc. | 0.928 | 0.928 | 1.000 | 0.971 | 1.000 | 0.990 | 0.522 |
|  | Top-1 Acc. | 0.760 | 0.760 | 1.000 | 0.963 | 1.000 | 0.984 | 0.427 |
| Readout | $\mathcal{L}_I/k$ | – | 2.500 | 1.822 | 0.628 | 0.222 | 0.079 | 0.023 |
|  | Top-5 Acc. | 0.928 | 0.928 | 0.930 | 0.928 | 0.917 | 0.870 | 0.505 |
|  | Top-1 Acc. | 0.760 | 0.760 | 0.761 | 0.756 | 0.735 | 0.660 | 0.302 |

Table 4: Influence of $\beta$ on the information loss $\mathcal{L}_I$ and the test accuracy on ResNet-50. $k$ is the size of the feature map, i.e. $k = hwc$. *Initial*: Configuration of the untrained bottleneck with $\alpha = 5$. *Original*: Values for the original model without the bottleneck.

| Parameter | ResNet-50 | VGG-16 | Search space |
|---|---|---|---|
| Target layer | conv2_3 | conv3_1 | |
| Reading out | conv2_3 | conv3_1 | |
|  | conv3_4 | conv4_1 | |
|  | conv4_6 | conv5_1 | |
|  | conv5_3 | conv5_3 | |
|  | fc | fc | |
| Optimizer | Adam (Kingma & Ba (2014)) | | |
| Learning Rate | $\eta = 10^{-5}$ | | {e-4, e-5, e-6} |
| Balance Factor | $\beta = 10/k$ | | $\{0.1/k, 1/k, 10/k, 100/k\}$ |
| Epochs | $E = 10$ | | |
| Batch Size | $B = 16$ | | |
| Smoothing | $\sigma_s = 1$ | | |

Table 3: Hyperparameters for the Readout Bottleneck.

In Table 2, we provide hyperparameters for the Per-Sample Bottleneck. In Table 3, we provide hyperparameters for the Readout Bottleneck.

In Table 4, a comparison of pre- to post-training accuracy and the estimated mutual information is shown for both Bottleneck types. The Per-Sample Bottleneck can learn to suppress negative evidence for each sample and the accuracy is close to 1 for $\beta \leq 10/k$. For $\beta = 100/k$, also positive evidence is discarded and the accuracy decreases to $0.43$. The Readout Bottleneck learns to suppress negative evidence for small $\beta \leq 0.1/k$ and slightly improves the final accuracy.

## G  EVALUATION METRICS

### G.1  SANITY CHECK: WEIGHT RANDOMIZATION

We use the cascading parameter randomization sanity check from Adebayo et al. (2018). Following the original paper, we used the skimage SSIM implementation with a window of size 5. For LRP, we found that the weight randomization flips the values of the saliceny heatmap, e.g. $h_r \approx -h_o$ where $h_r$ is the heatmap with random weights and $h_o$ the heatmap on the original model. Therefore for LRP, we used: $\max(\text{ssim}(h_o, \text{normalize}(h_r)), \text{ssim}(h_o, \text{normalize}(-h_r)))$. We normalized the heatmaps by first clamping the 1-th and 99-th percentile and then rescaling the heatmap it to $[0, 1]$. We run the sanity check on 200 randomly selected samples from the ImageNet valdiation set.

# H MoRF AND LeRF DEGRADATION PATHS

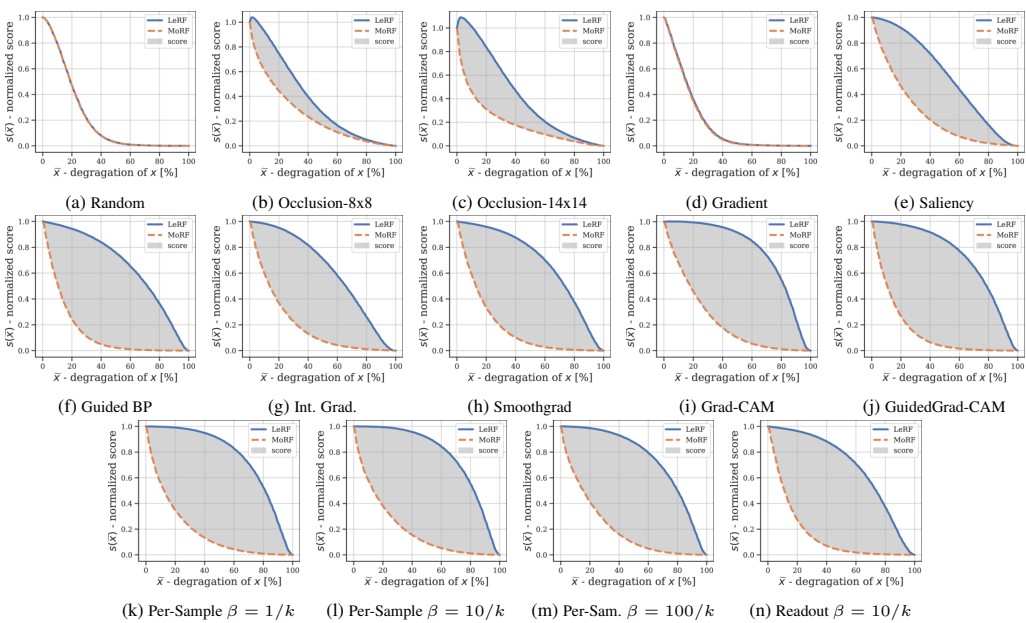

Figure 11: MoRF and LeRF for the ResNet-50 network using 14x14 tiles.

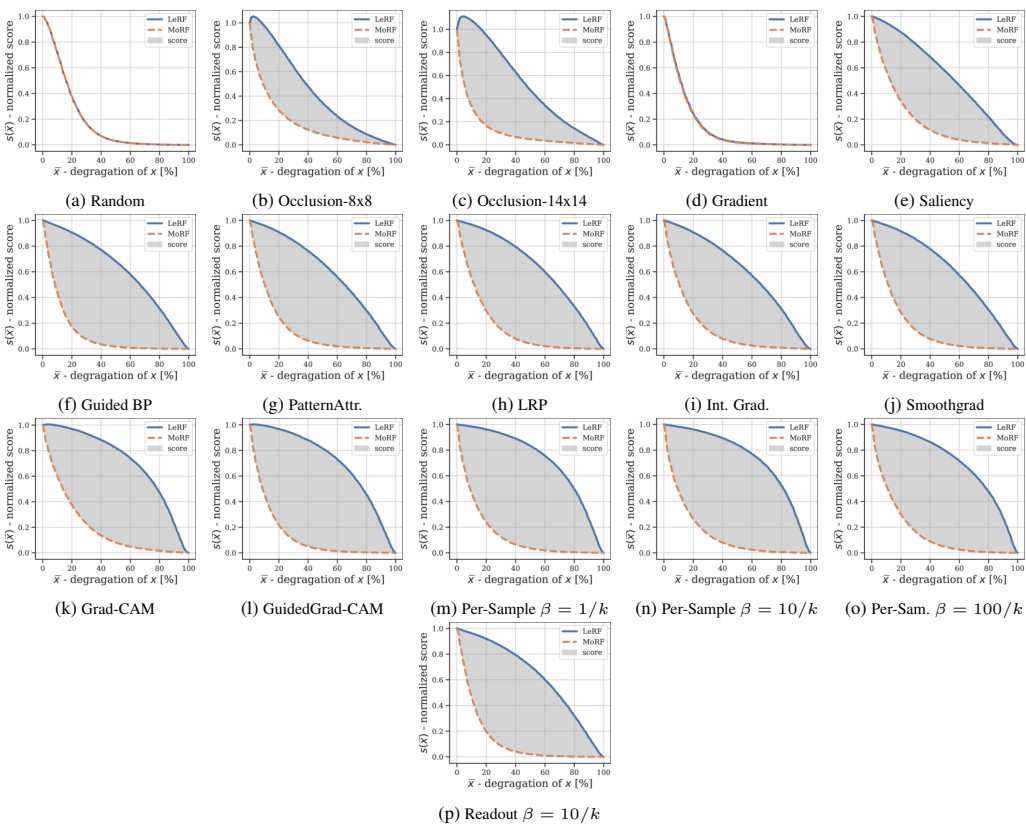

Figure 12: MoRF and LeRF paths for the VGG-16 network using 14x14 tiles.

