# OpenReview forum: "Restricting the Flow: Information Bottlenecks for Attribution"
_ICLR.cc/2020/Conference — Accept (Talk)_

### Official Review · AnonReviewer2 · 2019-10-22
**Official Blind Review #2**

**Rating:** 8

**Review:**

Summary
The paper proposes a novel perturbation-based method for computing attribution/saliency maps for deep neural network based image classifiers. In contrast to most previous work on perturbation-based attribution, the paper proposes to inject carefully crafted noise into an early layer of the network. Importantly, the noise is chosen such that it optimizes an information-theoretically motivated objective (rate-distortion/info bottleneck) that ensures that decision-relevant signal is flowing while constraining the overall channel-capacity, such that decision-irrelevant signal is blocked from flowing. The flow of signal is controlled by the amount of noise injected, which translates into a certain amount of mutual information between input image regions and noisy activations/features. This mutual information can be visualized in the input image, but it also has a clear, quantitative meaning that is readily interpretable. The paper introduces two ways to construct the injected noise, based on the information bottleneck. Resulting attribution maps are computed and evaluated on VGG-16 and ResNet-50 (on ImageNet), and are compared against an impressive number of previously proposed attribution methods. Importantly, the paper uses three different quantitative measures to compare the quality of attribution maps. The proposed method performs well on all three measures.

Contributions
i) Derivation of a novel method for constructing attribution maps. Importantly, the method is grounded on solid theoretical footing for extracting minimal relevant information (rate-distortion theory / information bottleneck method).

ii) Proposal of a novel quantitative measure to compare quality of pixel-level attribution maps in image classification, and extension of a previously reported method.

iii) Evaluation and comparison against a large body of state-of-the-art attribution methods.

Quality, Clarity, Novelty, Impact
The paper is clear and well written, with a nice introduction to the information bottleneck method. Experiments are well described and hyper-parameter settings are given in the appendix. To the best of my knowledge, the proposed method is sufficiently novel and the application of the information bottleneck framework to pixel-level attribution has not been reported before. Some of the design- and implementation-choices needed to render the intractable info bottleneck objective tractable could perhaps be discussed and potentially even improved in light of recent results in other fields (Bayesian DL, deep latent-variable generative models, and variational methods for deep neural network compression), but I currently don’t consider this a major issue. To me personally the work in convincing and mature enough to vote for acceptance - perhaps most importantly it lays important groundwork for important connections to the theory of relevant information and puts a lot of much needed emphasis on objective evaluation of attribution methods (i.e. without subjective visual judgement of saliency maps). My suggestions below are aimed at helping improve the paper even further.


Improvements
I) A short section of current shortcomings/limitations could be added to the discussion.

II) Perturbation-based approaches that inject noise (into the input image directly) have been proposed previously. Most notably: Visualizing and Understanding Atari Agents, Greydanus et al. 2018 and potentially follow-up citations. It would be interesting to compare both works empirically, but perhaps also theoretically/conceptually. Could the Greydanus work be related to applying the noise directly to the input image along with some additional constraints?


Minor Comments
a) Is there a particular reason for this choice of colormap? While it seems to be roughly perceptually uniform (which is of course good), why not choose a simple sequential colormap (instead of a rainbow-like one)? At least the use of red and green at the same time should rather be avoided to maximize colormap readability under the most common forms of color vision deficiencies.

b) Just a pointer - no need to act on this for the current paper. Large parts of the field of neural network compression are concerned with a similar kind of attribution - the question is which weights/neurons/filters are relevant and which ones are not and can thus be removed from the network without loss in accuracy. Information-bottleneck style objectives (or the closely related ELBO / variational free energy) in conjunction with sparsity inducing priors have been proven to be quite fruitful. See e.g. Variational Dropout Sparsifies Deep Neural Networks, Molchanov et al. 2017 for interesting work, that aims at learning the variance of Gaussian noise that is injected into neural network weights using a similar construction and variational objective as shown in this paper. Perhaps some ideas can be borrowed/translated for future, improved versions of the method from that body of literature (Molchanov 2017, but also more sophisticated follow-up work).

**Experience Assessment:**

I have published one or two papers in this area.

**Review Assessment: Checking Correctness Of Derivations And Theory:**

I assessed the sensibility of the derivations and theory.

**Review Assessment: Checking Correctness Of Experiments:**

I assessed the sensibility of the experiments.

**Review Assessment: Thoroughness In Paper Reading:**

I read the paper at least twice and used my best judgement in assessing the paper.

---

> ### Author Response · Authors · 2019-11-15
> **Response to Review 2**
>
> Thank you very much for your extensive and helpful comments. We addressed changes in the paper in a general comment. Concerning your specific comments:
>
> > [...]Some of the design- and implementation-choices needed to render the intractable info bottleneck objective tractable could perhaps be discussed and potentially even improved in light of recent results in other fields (Bayesian DL, deep latent-variable generative models, and variational methods for deep neural network compression),[...]
>
> Yes, a more complex variational approximation of Q(Z) could make our approximation of the mutual information even more accurate. As a simple normal distribution already yielded good results, we did not further explore this direction. However, it would be an interesting subject for future work.
>
> > A short section of current shortcomings/limitations could be added to the discussion.
>
> We agree! We added the following paragraph to the conclusion stating the following limitations:
>
> Generally, we would advise to use the Per-Sample Bottleneck over the Readout Bottleneck. It performs better and is more flexible as it only requires to estimate the mean and variance of the feature map. The Readout Bottleneck has the advantage of producing attribution maps with a single forward pass once trained. Images with multiple object instances provide the network with redundant class information. The Per-Sample Bottleneck may therefore discard some of the class evidence. Even for single object instances, the heatmaps of the Per-Sample Bottleneck may vary slightly due to the randomness of the optimization process.
>
>
> > II) Perturbation-based approaches that inject noise (into the input image directly) have been proposed previously. Most notably: Visualizing and Understanding Atari Agents, Greydanus et al. 2018 and potentially follow-up citations. It would be interesting to compare both works empirically, but perhaps also theoretically/conceptually. Could the Greydanus work be related to applying the noise directly to the input image along with some additional constraints?
>
> Thank you for the suggestion! Greydanus et al. blurs parts of the input images and then measures the drop in the output of the policy network and the value function. We think this method could be seen as an extension of Occlusion. Instead of setting image patches to zero, they are blurred, effectively removing high-frequency image information. Greydanus et. al do not apply noise to the input image and they also do not optimize the amount of blur. We cited the work as an Occlusion type method. We have searched the follow-up citations, but were not able to find any methods that apply noise for attribution purposes.
>
> > Is there a particular reason for this choice of colormap? While it seems to be roughly perceptually uniform (which is of course good), why not choose a simple sequential colormap (instead of a rainbow-like one)? At least the use of red and green at the same time should rather be avoided to maximize colormap readability under the most common forms of color vision deficiencies.
>
> We share your concerns and updated the colormap to red for positive attribution and blue for negative attribution.
>
> > Just a pointer - no need to act on this for the current paper. Large parts of the field of neural network compression are concerned with a similar kind of attribution - the question is which weights/neurons/filters are relevant and which ones are not and can thus be removed from the network without loss in accuracy. Information-bottleneck style objectives (or the closely related ELBO / variational free energy) in conjunction with sparsity inducing priors have been proven to be quite fruitful. See e.g. Variational Dropout Sparsifies Deep Neural Networks, Molchanov et al. 2017 for interesting work, that aims at learning the variance of Gaussian noise that is injected into neural network weights using a similar construction and variational objective as shown in this paper. Perhaps some ideas can be borrowed/translated for future, improved versions of the method from that body of literature (Molchanov 2017, but also more sophisticated follow-up work).
>
> Indeed, there exist interesting parallels to neural network compression. We agree that both areas could enrich each other. Thanks for pointing this out!

---

### Official Review · AnonReviewer3 · 2019-10-23
**Official Blind Review #3**

**Rating:** 8

**Review:**

This paper presents an information-bottleneck-based approach to infer the regions/pixels that are most relevant to the output. For all the metrics listed in the paper, the proposed approaches all achieve very good performance. It turns out, the proposed two architectures are better (at least alternative) choices to the other existing attribution methods.

I do agree that the proposed two models (Per-Sample and Readout) can be used to roughly infer regions of interest, which has been strongly supported by the comprehensive experiments. To minimize equation (6), we need to make beta*L_I small. Minimizing L_{CE} in (6) tries to maximize the mutual information between Z and output (labels); while minimizing L_I with respect to weight beta would try to inject noise to each dimension of Z. However, L_{CE} needs to ensure it can get enough information for prediction, and thus would prevent the noise injection process for “the key regions”. By choosing reasonable beta (similar to variational information bottleneck), the proposed approaches are capable to highlight key regions used for prediction.

Overall, I think the method is elegant for approximately estimating the relevance score map.
Below are some of my (minor) questions/concerns:

1. What we learned = What we want?
The proposed approach seeks a sort of “sparse heatmap”.
The larger the beta, the more regions/pixels would be suppressed while smaller beta might fail to suppress non-important regions in the image.
In the paper, the beta used for calculating the per-sample bottleneck is among [100/k , 10/k, 1/k].
The beta for ReadOut bottleneck is 10/k.
However, according to Table 1, only when beta is smaller than 1/k, the accuracy of the model does not degrade too much.
When using beta=10/k to get the "heat map" (where 10/k is the best choice of per-smaple bottleneck for degradation task), how close is the "heat map in beta=10/k" to the "ground-truth heatmap"?
To better understand the proposed methods, I have a small suggestion:
------ Try betas in a broader range including very small betas, e.g. [0.0001/k, 0.001/k,....,1/k,10/k], for both Table one and visualization.
Fix a few images and visualize the heatmap given different betas.
We might better see how the growth of beta changes the heatmap.

2. About zero-valued attributions.
I agree with you that equation (5) is an upper bound of MI (eq (4)).
However, I am not sure if I totally agree with the claim "If L_1 is zero for an area, we can guarantee that no information from this area is used for prediction."
----- Given L_1=0 really implies that no information of the corresponding region is used for the certain beta, but is this true for the original model (beta=0)? Table one shows that different beta would lead to very different downstream task accuracy.

3. Specific to the two approaches you proposed, can you explain/motivate in what situations per-sample bottle would be better and in what cases we should prefer ReadOut bottleneck?



**Experience Assessment:**

I have read many papers in this area.

**Review Assessment: Checking Correctness Of Derivations And Theory:**

N/A

**Review Assessment: Checking Correctness Of Experiments:**

I carefully checked the experiments.

**Review Assessment: Thoroughness In Paper Reading:**

I read the paper thoroughly.

---

> ### Author Response · Authors · 2019-11-15
> **Response to Review 3**
>
> Thank you very much for your comments. We addressed the majority of changes to the paper in a general response. Concerning your specific comments:
>
> > “How close is the "heat map in beta=10/k" to the "ground-truth heatmap"?”
>
> It is not clear to us what you mean by “ground-truth heatmap”. There is no human-labeled set of heatmaps available to evaluate attribution methods. To evaluate how well the attribution mass is localized, we used the „bbox“ metric which calculates the proportion of most relevant scores falling within the object‘s bounding box. Thus, we use the bounding box labels as ground-truth proxy for localization performance, and we find that beta=10/k performs best: For the ResNet-50, on average 62 % of the highest attribution values are contained in the respective bounding box which is 15.2% higher than the best baseline.
>
>
> > However, according to Table 1, only when beta is smaller than 1/k, the accuracy of the model does not degrade too much.
> > Try betas in a broader range including very small betas, e.g. [0.0001/k, 0.001/k,....,1/k,10/k], for both Table one and visualization.
>
> We agree with your suggestion and added a comparison of heatmaps for beta values from 0.1/k to 1000/k in a new figure (fig. 4). We found that beta = 0.1/k resulted in more information flowing through the network and producing more vague heatmaps. For beta = 1000/k, heatmaps are uniform with very low values (< 0.1 bits / pixel) meaning almost all information is discarded.
> We updated table 1 to also include the Per-Sample Bottleneck.
>
> > However, I am not sure if I totally agree with the claim "If L_1 is zero for an area, we can guarantee that no information from this area is used for prediction."
> ----- Given L_1=0 really implies that no information of the corresponding region is used for the certain beta, but is this true for the original model (beta=0)? Table one shows that different beta would lead to very different downstream task accuracy.
>
> We agree that that sentence could be clearer. In the introduction, we already described it clearer: “[..] areas scored irrelevant are indeed not necessary for the network's prediction.” We incorporated your feedback and changed the sentence to:  “If L_I is zero for an area, we can guarantee that information from this area is not necessary for the network's prediction. Information from this area might still be used when no noise is added.”
>
> > Specific to the two approaches you proposed, can you explain/motivate in what situations per-sample bottle would be better and in what cases we should prefer ReadOut bottleneck?
>
> We addressed the issues you raised, added additional content (stimulated by the other two reviewers) and hope you agree we have significantly improved the manuscript. We would appreciate if our efforts would be rewarded with an updated rating of “Accept”. Thank you!

---

### Official Review · AnonReviewer1 · 2019-10-25
**Official Blind Review #1**

**Rating:** 8

**Review:**


Summary
---

(motivation)
Lots of methods produce attribution maps (heat maps, saliency maps, visual explantions) that aim to highlight input regions with respect to a given CNN.
These methods produce scores that highlight regions that are in a vague sense "important."
While that's useful (relative importance is interesting), the scores don't mean anything by themselves.
This paper introduces another new attribution method that measures the amount of information (in bits!) each input region contains, calibrating this score by providing a reference point at 0 bits.
Non-highlighted regions contribute 0 bits of information to the task, so they are clearly irrelevant in the common sense that they have 0 mutual information with the correct output.

(approach - attribution methods)
An information bottleneck is introduced by replacing a layer's (e.g., conv2) output X with a noisy version Z of that output.
In particular, Z is a convex combination of the feature map (e.g., conv2) with Gaussian noise with the same mean and variance as that feature map.
The weights of the combination are found so they minimize the information shared between the input and Z and maxmimize information shared between Z and the task output Y.
These weights are either optimized on
1) a per-image basis (Per-Sample) or
2) predicted by a model trained on the entire dataset (Readout).

(approach - evaluation)
The paper uses 3 metrics with differing degrees of novelty:
1) The bbox metric rewards attribution methods that put a lot of mass in ground truth bounding boxes.
2) The original Sensitivity-n metric from (Ancona et al. 2017) is reported with a version that uses 8x8 occlusions.
3) Least relevant image degredation is compared to most relevant image degredation (e.g., from (Ancona et al. 2017)) to form a new occlusion style metric.

(experiments)
Experiments consider many of the most popular baselines, including Occlusion, Gradients, SmoothGrad, Integrated Gradients, GuidedBP, LRP, Grad-CAM, and Pattern Attribution. They show:
1) Qualitatively, the visualizations highlight only regions that seem relevant.
2) Both Per-Sample and Readout approaches put higher confidence into ground truth bounding boxes than all other baselines.
3) Both Per-Sample and Readout approaches outperform all baselines almost all the time according to the new image degredation metric.


Strengths
---

The idea makes a lot of sense. I think heat maps are often thought of in terms of the colloquial sense of information, so it makes sense to formalize that intuition.

The related work section is very well done. The first paragraph is particularly good because it gives not just a fairly comprehensive view of attribution methods, but also because it efficiently describes how they all work.

The results show that proposed approaches clearly outperform many strong baselines across different metrics most of the time.


Weaknesses
---


* I'm not sure why the new degredation metric is a useful addition. What does it add that MoRF and LeRF don't capture on their own independently?

* I think [1] would be a nice addition to the evaluation section as it tests for something qualitatively different than the various metrics from section 4. It would also be a good addition to the related work.


Missing Details / Points of Confusion
---

* I think there's an extra p(x) in eq. 11 in appendix D.

* I think the variable X is overloaded. In eq. 1 it refers to the input (e.g., the pixels of an image) while in eq. 2 it refers to an intermediate feature map (e.g., conv2) even though it later seems to refer to the input again (e.g., eq. 3). Different notation should be used for intermediate feature maps and inputs.


Presentation Weaknesses
---

* In section 3.1 is lambda meant to be constrained in the range [0, 1]? This is only mentioned later (section 3.2) and should probably be mentioned when lambda is introduced.

* "indicating that all negative evidence was removed." I think this should read "indicating that only negative evidence was removed."


Suggestions
---

"The bottleneck is inserted into an early layer to ensure that the information in the network is still local"
I'd like this to be explored a bit more. Though deeper feature maps are certainly more spatially coarse they still might be somewhat "local". To what degree to they loose localization information? My equally vague alternative intuition goes a bit differently: The amount of relevant information flowing through any spatial location seems like it shouldn't change that much, only the way its represented should change. If the proposed visualizations were the same for every choice of layer then it would confirm this intuition. That would also be an interesting result because most if not all of the cited baseline approaches (where applicable) produce qualitatively different attributions at different layers (e.g., see Grad-CAM).


[1]: Adebayo, Julius et al. “Sanity Checks for Saliency Maps.” NeurIPS (2018).


Preliminary Evaluation
---

Clarity: The paper is clearly written.
Originality: The idea of using the formal notion of information in attribution maps is novel, as is the bbox metric.
Significance: This method could be quite significant. I can see it becoming an important method to compare to.
Quality: The idea is sound and the evaluation is strong.

This is a very nice paper in all the ways listed above and it should be accepted!

Post-rebuttal comments
---

The author responses and other reviews have only increased my confidence that this paper should be accepted.



**Experience Assessment:**

I have published one or two papers in this area.

**Review Assessment: Checking Correctness Of Derivations And Theory:**

N/A

**Review Assessment: Checking Correctness Of Experiments:**

I assessed the sensibility of the experiments.

**Review Assessment: Thoroughness In Paper Reading:**

I read the paper at least twice and used my best judgement in assessing the paper.

---

> ### Author Response · Authors · 2019-11-15
> **Response to Review 1**
>
> Thank you for your extensive and helpful comments and thorough review of the paper. We summarized our modifications in a general comment. We respond inline to your specific comments:
>
> > I'm not sure why the new degradation metric is a useful addition. What does it add that MoRF and LeRF don't capture on their own independently?
>
> We agree that the integral between MoRF and LeRF does not capture anything not already implicitly contained in the MoRF and LeRF curves. However, when comparing different MoRF or LeRF curves visually, it is not always obvious which method performs better overall, as the paths may intersect (see Appendix G). Calculating the integral between the MoRF and LeRF paths yields a single scalar, which is directly comparable and while capturing the objective to perform well in both the MoRF and LeRF task.
>
> > I think [1] would be a nice addition to the evaluation section as it tests for something qualitatively different than the various metrics from section 4. It would also be a good addition to the related work.
>
> Thanks for pointing it out to us. We added the weight randomization sanity check [1] to the evaluation section and compare our method to the others.
>
> Regarding your minor comments / presentation issues:
> * we removed the p(x) in eq. 11.
> * we now mention the range of lambda when it is introduced
> * we introduced a new variable R to denote intermediate feature maps
>
> > "indicating that all negative evidence was removed." I think this should read "indicating that only negative evidence was removed."
>
> Thank you, we updated the paper accordingly.
>
> > "The bottleneck is inserted into an early layer to ensure that the information in the network is still local". I'd like this to be explored a bit more. Though deeper feature maps are certainly more spatially coarse they still might be somewhat "local". To what degree to they loose localization information? My equally vague alternative intuition goes a bit differently: The amount of relevant information flowing through any spatial location seems like it shouldn't change that much, only the way its represented should change. If the proposed visualizations were the same for every choice of layer then it would confirm this intuition. That would also be an interesting result because most if not all of the cited baseline approaches (where applicable) produce qualitatively different attributions at different layers (e.g., see Grad-CAM).
>
> This is indeed an interesting question and we included a new figure which compares different layer depths. The figure backs your intuition that the spatial locations of important features should remain approximately the same. However, deeper layers have larger FOVs, so that the representations are not guaranteed to stay in the exact spatial location. Deeper layers also have drastically smaller spatial resolution, limiting the resolution of the heatmap. For very early layers, the heatmaps are sparser.

---

### Author Response · Authors · 2019-10-15
**bug with minor effects on the results**

Hello,

we found a bug in our code that had minor effects on our results.  When calculating the KL-divergence, we used "log(s)" instead of "log(s**2)" where "s" is the standard deviation. We re-run the evaluation and provide an screenshot of the updated results: https://gist.github.com/attribution-bottleneck/07ee0959bbd8b8ac36f9dba476301dd8 .
The degradation task for VGG is now also performed on the full ImageNet validation set and on 8x8 and 14x14 tiles. Using the correct log variance, we found that the Per-Sample bottleneck even improved a bit on the degradation task. We will update the paper once the rebuttal period starts.

---

### Author Response · Authors · 2019-11-15
**General Response to the Reviews**

We want to thank the reviewers for the extensive feedback, their helpful comments, and suggestions. We appreciate the effort and time you invested very much! We respond to each review below individually. Here is a short summary of our improvements:

* A new figure shows the effect of varying layer depth and varying values for beta
* included Per-Sample Bottleneck to Table 1
* we added "sanity checks" (Adebayo et. al, 2018) to our evaluation section
* we fixed typos, integrated minor comments, and improved the presentation

---

### Public Comment · ~Mark_Sandler1 · 2020-01-08
**Some related work...**

Nice paper!

Also See: Information-Bottleneck Approach to Salient Region Discovery by Zhmoginov et all,
https://arxiv.org/abs/1907.09578 also explores information bottleneck for similar tasks on simple datasets.

Their (our) model is somewhat different, but relies on similar concept of finding the regions that preserve most of the mutual information between masked image and the labels. It would be interesting if the differences were articulated in this paper.

---

### Author Response · Authors · 2020-02-15
**Summary of Changes**

We want to summarize our changes since the original submission:
- Include Sanity Checks (Adebayo et al., 2018)
- Add Figure 4: Different depth and beta values
- Include LRP with parameters α=1, β=0
- Include some additional references

Improved presentation:
- switch to seismic color map.
- migrate the overloading of X by introducing a new variable for the intermediate representation R.
- include heatmaps in the appendix without overlay on not-cherry-picked samples.
- redo diagrams of Per-Sample and Readout with tikz (increase beauty).
- fixed several minor issues (grammar, wording, clarity).

Many of these changes were encouraged by the feedback of our reviewers.
Thank you for accepting us for a talk!

---

### Public Comment · ~Saeid_Asgari_Taghanaki1 · 2020-03-30
**Overlaps with the Infomask paper**

Dear authors, we found overlaps in the methodology of your paper with our published Infomask paper: https://arxiv.org/abs/1903.11741

It is highly appreciated if you could please highlight the differences.

Thanks

---

> ### Author Response · Authors · 2020-05-07
> **Infomask paper**
>
> Dear Saeid,
>
> thanks for pointing us to your paper. We now reference your work. The main difference is that in your work the information bottleneck is already added during the training of the network. In contrast, our methods works aims at already trained network (post-hoc explanations). This is also reflected how we restrict the amount of information.
>
> Best,
> Leon

---

### Decision · Program_Chairs · 2019-12-19

**Decision:**

Accept (Talk)

**Comment:**

All three reviewers strongly recommend accepting this paper. It is clear, novel, and a significant contribution to the field. Please take their suggestions into account in a camera ready version. Thanks!